# Secrecy Outage Probability of a NOMA Scheme and Impact Imperfect Channel State Information in Underlay Cooperative Cognitive Networks

**DOI:** 10.3390/s20030895

**Published:** 2020-02-07

**Authors:** Tan-Phuoc Huynh, Duy-Hung Ha, Cong Truong Thanh, Peppino Fazio, Miroslav Voznak

**Affiliations:** 1Department of Telecommunications, VSB-Technical University of Ostrava, 17. listopadu 2172/15, 708 00 Ostrava, Czech Republic; htphuocsuna@gmail.com (T.-P.H.); duy.hung.ha.st@vsb.cz (D.-H.H.); cong.thanh.truong.st@vsb.cz (C.T.T.); peppino.fazio@vsb.cz (P.F.); 2Department of Computer Networks and Data Communications, Eastern International University (EIU), Nam Ky Khoi Nghia, Binh Duong 75000, Vietnam

**Keywords:** non-orthogonal multiple access, physical layer security (PLS), cooperative communication, successive interference cancellation (SIC), decode-and-forward (DF), cognitive radio (CR), channel state information, outage probability

## Abstract

Security performance and the impact of imperfect channel state information (CSI) in underlay cooperative cognitive networks (UCCN) is investigated in this paper. In the proposed scheme, relay R uses non-orthogonal multiple access (NOMA) technology to transfer messages $e_{1}$, $e_{2}$ from the source node S to User 1 (U_1_) and User 2 (U_2_), respectively. An eavesdropper (E) is also proposed to wiretap the messages of U_1_ and U_2_. The transmission’s security performance in the proposed system was analyzed and performed over Rayleigh fading channels. Through numerical analysis, the results showed that the proposed system’s secrecy performance became more efficient when the eavesdropper node E was farther away from the source node S and the intermediate cooperative relay R. The secrecy performance of U_1_ was also compared to the secrecy performance of U_2_. Finally, the simulation results matched the Monte Carlo simulations well.

## 1. Introduction

The UCCN is known as the CR which is a promising technology and innovative solution for dealing with the radio spectrum allocation and precise requirements issues [1]. CR permits secondary users (SUs or unlicensed users) to access the dormant frequency spectrum without causing interruption to the primary users (PUs or licensed user). Due to SUs being accepted to the PUs at the same time, the SUs have to keep their transmit powers within the acceptable levels. Besides that, with rapidly extending wireless sensor networks (WSNs) in many areas of industry, the security of information transfer becomes a more serious problem. Many researchers investigated PLS to help security transmission between the source node and the destination node to improve and enhance the secrecy of WSNs.

Recently, many solutions and technologies have been investigated for the purposes of speeding up mobile data transmission, extending wireless communication range, and assisting users in connecting security together. Examples of these technologies include amplify-and-forward (AF), orthogonal multiple access (OMA), and energy harvesting [2,3,4]. NOMA technology, however, is a promising method and has attracted significant attention in recent years [5,6,7,8,9,10].

NOMA technology has gradually become one of the most efficient solutions in developing the fifth-generation mobile network (5G). In the NOMA technique, the users can share both time and frequency resources and only adjust their power allocation ratios. The users with better channel conditions can serve as relays to enhance the system performance by using SIC [9]. This technology improves the limitation of orthogonal multiple access (OMA). It meets the needs of end users in providing access to data quickly and securely. NOMA and PLS are therefore very important techniques in data transfer. They assist in transmitting signals from the source node to destination node with high speed, efficiency, and data confidentiality.

Several studies have examined NOMA and PLS in wireless systems [11,12,13]. In [11], the authors considered a cooperative relaying system using the NOMA technique to enhance the efficiency of the transmitted signal. The researchers in [13] investigated the effectiveness of new schemes that combined partial relay selection and NOMA in AF relaying systems to increase data transmission rates for 5G mobile networks.

A considerable amount of literature has been published on PLS [14,15,16]. In [14], the authors analyzed the secrecy performance of cooperative protocols with relay selection methods influenced by co-channel interference. The authors in [15] inspected the impact of correlated fading on the secrecy performance of multiple DF relaying that uses the optimal relay selection method. Some researchers have also combined the NOMA technique with PLS [17,18,19]. In [17], the authors resolved the problem of maximizing the minimum confidential information rate in users subject to the secrecy outage constraint and instantaneous transmit power constraint. Cooperative NOMA systems with PLS in both AF and DF were studied by the authors in [18].

The application of NOMA techniques and security principles in underlay cognitive radio networks were also suggested by some authors in [20,21,22,23,24]. In [20], the authors discussed a cooperative transmission scheme for a downlink NOMA in CR systems. This research exploited maximum spatial diversity. The researchers in [24] considered secure communication in cognitive DF relay networks in which a pair of cognitive relays were opportunistically selected for security protection against eavesdroppers.

Channel state information (CSI) has a vital role in wireless communication systems. It describes how a signal propagates from the source node to the relay, such as scattering, fading, and power decay over distance. During a receiver’s set-up period, the CSI is evaluated and transferred to related nodes in the system through a media access control protocol. In [25], the authors researched the effect of imperfect channel CSI on secondary users in an underlay DF cognitive network with multiple primary receivers. In [26], scientists studied the effect of imperfect CSI on a DF cooperative underlay cognitive radio NOMA network in order to determine the optimal power allocation factors for different user distances.

In most of the literature reported above, the combination of NOMA and PLS in a UCCN influenced by CSI was not proposed. Motivated and inspired by the above ideas, a cooperative scheme is suggested in this paper. In this scheme, a proposed UCCN using NOMA is required to both decode and forward the messages $e_{1}$ and $e_{2}$ from node S to two destination nodes (U${}_{1}$ and U${}_{2}$) under the effect of CSI and an eavesdropper. The secrecy performance of the communications $e_{1}$ and $e_{2}$ in the proposed system were then examined and estimated in terms of secrecy outage probability over Rayleigh fading channels to improve spectral efficiency and secure communication.

The main contributions of the paper are summarized as follows:-A study of the impact of imperfect CSI and the secrecy performance of a UCCN applying the NOMA technique to improve system performance in a 5G wireless network.-Secrecy outage probability (SOP) is performed over Rayleigh fading channels and verified with Monte Carlo simulations.-The results achieved by the proposed scheme demonstrate the security performance of U${}_{1}$ and U${}_{2}$.-The secrecy performance of the proposed system improved when the distance between the eavesdropper node E and the source and cooperative relay increased.

The paper has five sections. Section 1 introduces the topic. Section 2 describes the proposed scheme’s system model. Section 3 presents the results of an analysis of the secrecy outage probability at the source nodes. Section 4 presents the simulation results. Section 5 summarizes the conclusions.

## 2. System Model

Figure 1 illustrates the influence of imperfect CSI in a UCCN using NOMA and PLS. The system model consists of the source nodes S transferring a superimposed signal $e_{1}$ and $e_{2}$ to U${}_{1}$ and U${}_{2}$, respectively, through relay node R. One eavesdropper node E is proposed to wiretap the signals $e_{1}$, $e_{2}$ of the links S-U${}_{1}$, S-U${}_{2}$. In addition, the system model also consists of a node Pu which is known as the primary user having the license. Due to the interference constraint at the Pu node in the UCCN, the relay R and source S adjust their transmitting powers. In this model, we assume that the intermediate relay node R operates in DF relaying method and applies the NOMA principle under the influence of imperfect CSIs and PLS in UCCN. In addition, the variances of Zero-mean White Gaussian Noises (AWGNs) are equal, given as $N_{0}$. In this work, the corresponding distances of the links S-Pu, R-Pu, S-R, S-E, R-E, R-Pu, R-U${}_{1}$, and R-U${}_{2}$ in Figure 1 are given as $l_{SPu},l_{RPu},l_{SR},l_{SE},l_{RE},l_{RPu},l_{1}$, and $l_{2}$.

Regarding the system channels, $h_{i}$ represents the Rayleigh fading channel coefficient, $i\in \left(h_{SR},\;h_{SE},h_{SPu},h_{RE},h_{RPu},1,2\right)$. We assume that the channels $h_{i}$ do not change during block time T and are independently and identically distributed between two consecutive block times [10].

Finally, all of nodes in the system model have a single antenna for transmitting and receiving messages.

In principle, there are two time slots involved in each system communication process, and are given as follows:

At the first time slot, the source node S transfers the information e${}_{S}$ to the relay R and the eavesdropper node E, which is given by the math expression as
(1)$$e_{s}=\sqrt{\beta_{1}P_{s}}e_{1}+\sqrt{\beta_{2}P_{s}}e_{2},$$
where Ps is the power at source node S, $e_{1}$, and $e_{2}$ are the messages of U${}_{1}$, U${}_{2}$, respectively, with $E\left\{|e_{j}{|}^{2}\right\}=1,\;\;j\in \left(1,2\right)$, (E{e} being notated for the expectation process of *e*). The $\beta_{1}$ and $\beta_{2}$ are the power allocation coefficients. Following the principle of the NOMA, we assume that $\beta_{1}>\beta_{2}$ with $\beta_{1}+\beta_{2}=1$.

Because of the estimation errors of channels $h_{i}$, the evaluated fading channel coefficients at the nodes are represented as follows [25]:(2)$$\hat{h_{i}}=\rho h_{i}+\sqrt{1-\rho^{2}}\varepsilon_{i},$$
where $\hat{h_{i}},\;h_{i},\;\;\mathrm{and}\;\;\varepsilon_{i}$ are modeled as the additive white Gaussian noise (AWGN) with the random variable $f_{i}=|\hat{h_{i}}{|}^{2}$. The correlation coefficient $\rho \in \left[0,1\right]$ is described as the average quality of the channel estimation.

Notation: The Cumulative Distribution Function (CDF) and probability density function (pdf) of the random variable $f_{i}$ is denoted respectively as $F_{f_{i}}\left(x\right)=1-e^{-\frac{1}{\lambda_{i}}x}$ and $f_{f_{i}}\left(x\right)=\frac{1}{\lambda_{i}}e^{-\frac{1}{\lambda_{i}}x}$, where $\lambda_{i}=l_{i}^{-\beta }$, and β is a path-loss exponent.

The received signal at R from source node S for decode $e_{1}$ under impact imperfect CSIs is given as follows:
(3a)$$\begin{array}{c} y_{SR}^{e_{1}}=h_{SR}e_{s}+\sigma_{R}. \end{array}$$

Replace $h_{SR}$ from formula (2), the signal $y_{SR}^{e_{1}}$ is calculated as
(3b)$$\begin{array}{cc} y_{SR}^{e_{1}} & =\sqrt{\beta_{1}P_{s}}e_{1}\left(\frac{{\hat{h}}_{SR}-\sqrt{1-\rho^{2}}\varepsilon_{SR}}{\rho }\right)+\sqrt{\beta_{2}P_{s}}e_{2}\left(\frac{{\hat{h}}_{SR}-\sqrt{1-\rho^{2}}\varepsilon_{SR}}{\rho }\right)+n_{R}\\ & =\frac{\sqrt{\beta_{1}P_{s}}e_{1}{\hat{h}}_{SR}}{\rho }+\frac{\sqrt{\beta_{2}P_{s}}e_{2}{\hat{h}}_{SR}}{\rho }-\frac{\sqrt{\beta_{1}P_{s}}e_{1}\sqrt{1-\rho^{2}}\varepsilon_{SR}}{\rho }-\frac{\sqrt{\beta_{2}P_{s}}e_{2}\sqrt{1-\rho^{2}}\varepsilon_{SR}}{\rho }+n_{R}, \end{array}$$
where $n_{R}$ denotes the AWGNs at the relay R with the same variance $N_{0}$.

Because of applying NOMA technology, thanks to the deployment of SIC in NOMA principle, firstly, the relay R decodes the signal $e_{1}$ from formula (3b) and removes it, then the signal $e_{2}$ will be decoded without the component $\frac{\sqrt{\beta_{1}P_{s}}e_{1}{\hat{h}}_{SR}}{\rho }$ in formula (3b). Therefore, the signal $e_{2}$ received at R from source S after removing the signal $e_{1}$ is expressed as follows:(4)$$y_{SR}^{e_{2}}=\frac{\sqrt{\beta_{2}P_{s}}e_{2}{\hat{h}}_{SR}}{\rho }-\frac{\sqrt{\beta_{1}P_{s}}e_{1}\sqrt{1-\rho^{2}}\varepsilon_{SR}}{\rho }-\frac{\sqrt{\beta_{2}P_{s}}e_{2}\sqrt{1-\rho^{2}}\varepsilon_{SR}}{\rho }+n_{R}.$$

Similarly, the node E also wiretaps the packets $e_{1}$ and $e_{2}$ from S, respectively, and the received signals at node E are obtained as follows:(5)$$y_{SE}^{e_{1}}=\frac{\sqrt{\beta_{1}P_{s}}e_{1}{\hat{h}}_{SE}}{\rho }+\frac{\sqrt{\beta_{2}P_{s}}e_{2}{\hat{h}}_{SE}}{\rho }-\frac{\sqrt{\beta_{1}P_{s}}e_{1}\sqrt{1-\rho^{2}}\varepsilon_{SE}}{\rho }-\frac{\sqrt{\beta_{2}P_{s}}e_{2}\sqrt{1-\rho^{2}}\varepsilon_{SE}}{\rho }+n_{E}$$
(6)$$y_{SE}^{e_{2}}=\frac{\sqrt{\beta_{2}P_{s}}e_{2}{\hat{h}}_{SE}}{\rho }-\frac{\sqrt{\beta_{1}P_{s}}e_{1}\sqrt{1-\rho^{2}}\varepsilon_{SE}}{\rho }-\frac{\sqrt{\beta_{2}P_{s}}e_{2}\sqrt{1-\rho^{2}}\varepsilon_{SE}}{\rho }+n_{E},$$
where $n_{E}$ denotes the AWGNs at the E with the same variance $N_{0}$.

In the second time slot, after the received signals, the relay R sends them to the source nodes U${}_{1}$ and U${}_{2}$. Hence, the received signals at the destination node U${}_{1}$, U${}_{2}$ are given respectively as
(7)$$y_{RU_{1}}^{e_{1}}=\frac{\sqrt{\beta_{1}P_{R}}e_{1}\hat{h_{1}}}{\rho }+\frac{\sqrt{\beta_{2}P_{R}}e_{2}\hat{h_{1}}}{\rho }-\frac{\sqrt{\beta_{1}P_{R}}e_{1}\sqrt{1-\rho^{2}}\varepsilon_{1}}{\rho }-\frac{\sqrt{\beta_{2}P_{R}}e_{2}\sqrt{1-\rho^{2}}\varepsilon_{1}}{\rho }+n_{U_{1}}$$
(8)$$y_{RU_{2}}^{e_{2}}=\frac{\sqrt{\beta_{2}P_{R}}e_{2}\hat{h_{2}}}{\rho }-\frac{\sqrt{\beta_{1}P_{R}}e_{1}\sqrt{1-\rho^{2}}\varepsilon_{2}}{\rho }-\frac{\sqrt{\beta_{2}P_{R}}e_{2}\sqrt{1-\rho^{2}}\varepsilon_{2}}{\rho }+n_{U_{2}},$$
where $n_{U_{1}},\;n_{U_{2}}$ denote the AWGNs at the destination U${}_{1}$, U${}_{2}$ with the same variance $N_{0}$, and $P_{R}$ is a transmit power of the relay R.

In the proposed scheme, under the interference constraint at the node *P_u_*, the source node S and relay node R have to adjust their transmitting powers so that the interference power at the Pu must be less than a threshold value, which is assumed as *I*${}_{th}$. The maximum powers of nodes S and R are given, respectively,
(9a)$$\begin{array}{c} P_{S}=\frac{I_{th}}{|\hat{h_{SR}}{|}^{2}}=\frac{I_{th}}{f_{SR}}. \end{array}$$
(9b)$$\begin{array}{c} P_{R}=\frac{I_{th}}{|\hat{h_{RPu}}{|}^{2}}=\frac{I_{th}}{f_{RPu}}. \end{array}$$

Because the node E connects to the relay R directly, so it wiretaps the packets $e_{1}$ and $e_{2}$ from relay R. Therefore, the received signals at E through the link R-E are expressed as
(10)$$y_{RE}^{e_{1}}=\frac{\sqrt{\beta_{1}P_{R}}e_{1}{\hat{h}}_{RE}}{\rho }+\frac{\sqrt{\beta_{2}P_{R}}e_{2}{\hat{h}}_{RE}}{\rho }-\frac{\sqrt{\beta_{1}P_{R}}e_{1}\sqrt{1-\rho^{2}}\varepsilon_{RE}}{\rho }-\frac{\sqrt{\beta_{2}P_{R}}e_{2}\sqrt{1-\rho^{2}}\varepsilon_{RE}}{\rho }+n_{E}.$$
(11)$$y_{RE}^{e_{2}}=\frac{\sqrt{\beta_{2}P_{R}}e_{2}{\hat{h}}_{RE}}{\rho }-\frac{\sqrt{\beta_{1}P_{R}}e_{1}\sqrt{1-\rho^{2}}\varepsilon_{RE}}{\rho }-\frac{\sqrt{\beta_{2}P_{R}}e_{2}\sqrt{1-\rho^{2}}\varepsilon_{RE}}{\rho }+n_{E}.$$

We define the received Signal-to-Interference and Noise Ratios (SINRs) as γ=E{|signal|2}E{|overall noise|^2^}.

Firstly, we calculate the received Signal-to-Interference and Noise Ratios (SINRs) for decoding the information signal e${}_{1}$.

Thus, from formula (3b), the SINR at the relay R with the link S-R is obtained as follows:
(12a)$$\begin{array}{ll} \gamma_{SR}^{e_{1}} & =\frac{\frac{\beta_{1}P_{S}|\hat{h_{SR}}{|}^{2}}{\rho^{2}}}{\frac{\beta_{2}P_{S}|\hat{h_{SR}}{|}^{2}}{\rho^{2}}+\frac{\beta_{1}P_{S}\left(1-\rho^{2}\right)\lambda_{SR}}{\rho^{2}}+\frac{\beta_{2}P_{S}\left(1-\rho^{2}\right)\lambda_{SR}}{\rho^{2}}+N_{0}}\\ & =\frac{\beta_{1}P_{S}f_{SR}}{\beta_{2}P_{S}f_{SR}+P_{S}\left(1-\rho^{2}\right)\lambda_{SR}\left(\beta_{1}+\beta_{2}\right)+\rho^{2}N_{0}}. \end{array}$$

Replacing $P_{S}=\frac{I_{th}}{f_{SR}}$ in (9a), and setting $P=\frac{I_{th}}{N_{0}}$, $\gamma_{SR}^{e_{1}}$ is rewritten as
(12b)$$\begin{array}{c} \gamma_{SR}^{e_{1}}=\frac{P\beta_{1}f_{SR}}{P\beta_{2}f_{SR}+P\left(1-\rho^{2}\right)\lambda_{SR}+\rho^{2}f_{SPu}}, \end{array}$$

Similarly, with the formula in (7), we also calculate $\gamma_{RU_{1}}^{e_{1}}$, and this is achieved by mathematical expression as
(13)$$\gamma_{RU_{1}}^{e_{1}}=\frac{P\beta_{1}f_{1}}{P\beta_{2}f_{1}+P\left(1-\rho^{2}\right)\lambda_{1}+\rho^{2}f_{RPu}},$$
where $P=\frac{I_{th}}{N_{0}}$.

Applying formulas (5) and (10), the received SINRs at the eavesdropper node E with the link S-E and R-E are given, respectively, as follows:(14)$$\begin{array}{cc} \gamma_{SE}^{e_{1}} & =\frac{\beta_{1}P_{s}f_{SE}}{\beta_{2}P_{s}f_{SE}+P_{s}\left(1-\rho^{2}\right)\lambda_{SE}+\rho^{2}N_{0}}\\ \; & =\frac{P\beta_{1}f_{SE}}{P\beta_{2}f_{SE}+P\left(1-\rho^{2}\right)\lambda_{SE}+\rho^{2}f_{SPu}}. \end{array}$$
(15)$$\gamma_{RE}^{e_{1}}=\frac{P\beta_{1}f_{RE}}{P\beta_{2}f_{RE}+P\left(1-\rho^{2}\right)\lambda_{RE}+\rho^{2}f_{RPu}}.$$

In the second, similar to decoding the information signal $e_{1}$, we find the received SINRs for decoding the information signal $e_{2}$ as follows.

We apply formulas (4) and (6), the received SINRs at the nodes R with the link S-R, and at the eavesdropper node E with the link S-E are expressed, respectively, as follows:(16)$$\begin{array}{cc} \gamma_{SR}^{e_{2}} & =\frac{\beta_{2}P_{s}f_{SR}}{\beta_{1}P_{s}\left(1-\rho^{2}\right)\lambda_{SR}+\beta_{2}P_{s}\left(1-\rho^{2}\right)\lambda_{SR}+\rho^{2}N_{0}}=\frac{P\beta_{2}f_{SR}}{P\left(1-\rho^{2}\right)\lambda_{SR}+\rho^{2}f_{SPu}}. \end{array}$$
(17)$$\begin{array}{cc} \gamma_{SE}^{e_{2}} & =\frac{\beta_{2}P_{s}f_{SE}}{P_{s}\left(1-\rho^{2}\right)\lambda_{SE}+\rho^{2}N_{0}}=\frac{P\beta_{2}f_{SE}}{P\left(1-\rho^{2}\right)\lambda_{SE}+\rho^{2}f_{SPu}}. \end{array}$$

Similarly, with formulas (8) and (12), the received SINRs at the nodes U${}_{2}$ and E from relay R are inferred, respectively, as follows:(18)$$\gamma_{RU_{2}}^{e_{2}}=\frac{P\beta_{2}f_{2}}{P\left(1-\rho^{2}\right)\lambda_{2}+\rho^{2}f_{RPu}}.$$
(19)$$\gamma_{RE}^{e_{2}}=\frac{P\beta_{2}f_{RE}}{P\left(1-\rho^{2}\right)\lambda_{RE}+\rho^{2}f_{RPu}}.$$

Applying the Shannon capacity formula, the achievable rates of the links *X–Y* are formulated as
(20)$$R_{XY}^{e_{j}}=\frac{1}{2}\log_{2}\left(1+\gamma_{{}_{XY}}^{e_{j}}\right).$$
where the ratio 1/2 represents the fact that data transmission is split into two time slots, $X\in \left\{S,R\right\}$, and $Y\in \left\{E,U_{1},U_{2}\right\}$. The secrecy capacity of the UCCN systems with DF-based NOMA for the S-U${}_{j}$ communication can be expressed as
(21)$$SC_{j}={\left[SC_{U_{j}}^{e_{j}}-SC_{E}^{e_{j}}\right]}^{+},$$
where ${\left[x\right]}^{+}=\max\left(0,x\right)$; $SC_{SR}^{e_{i}}$ and $SC_{RUj}^{e_{j}}$ are the secrecy capacities from the source node S to the relay R and from the relay R to the destination U${}_{i}$ are given, respectively, as
(22)$$SC_{SR}^{e_{j}}=\max\left(0,R_{SR}^{e_{j}}-R_{SE}^{e_{j}}\right).$$
(23)$$SC_{RU_{i}}^{e_{j}}=\max\left(0,R_{RU_{i}}^{e_{j}}-R_{RE}^{e_{j}}\right).$$

## 3. Secrecy Outage Probability Analysis

In this section, the secrecy outage probability for eavesdropping the signals of U${}_{1}$ and U${}_{2}$ in the proposed scheme are analyzed. We assume that a node successfully and safely decodes the received packet if its achievable secrecy capacity is larger than a threshold secrecy capacity $SC_{th}$.

### 3.1. Secrecy Outage Probability of U_1_

The secrecy outage probability of U${}_{1}$ occurring when U${}_{1}$ does not receive a signal safely from the source node S under the malicious attempt of the eavesdropper E is expressed as follows:(24)$$\begin{array}{cc} OP_{U_{1}} & =\mathrm{Pr}[min\left(SC_{SR}^{e_{1}},SC_{RU_{1}}^{e_{1}}\right)<SC_{th}]\\ & =1-\mathrm{Pr}[SC_{SR}^{e_{1}}\geq SC_{th},SC_{RU_{1}}^{e_{1}}\geq SC_{th}]. \end{array}$$

Replacing $SC_{SR}^{e_{1}}=\max\left(0,R_{SR}^{e_{1}}-R_{SE}^{e_{1}}\right)$ at formula (22) and $SC_{RU_{1}}^{e_{1}}=\max\left(0,R_{RU_{1}}^{e_{1}}-R_{RE}^{e_{1}}\right)$ at (23) in formula (24), the $OP_{U_{1}}$ is rewritten as follows:(25)$$OP_{U_{1}}=1-\underset{\mathrm{Pr}1.1}{\underset{︸}{\mathrm{Pr}\left[R_{SR}^{e_{1}}-R_{SE}^{e_{1}}\geq SC_{th}\right]}}\times \underset{\mathrm{Pr}1.2}{\underset{︸}{\mathrm{Pr}\left[R_{RU_{1}}^{e_{1}}-R_{RE}^{e_{1}}\geq SC_{th}\right]}}$$

**Proposition** **1.**
*The probability of the Pr1.1, and Pr1.2 in (25) is given as*
(26)Pr1.1=0a≤θb1/λSPue−ψ11/λSR1/λSPu+ψ2/λSR−11λSRλSPuλSRλSPuI1a>θb.
*where*
$$\begin{array}{l} \psi 11=\frac{\phi c\lambda_{SE}}{\left(a-\phi b\right)};\;\psi 2=\frac{\phi \rho^{2}}{\left(a-\phi b\right)}\\ I_{1}=\int_{0}^{\infty }\int_{\left(\psi_{11}+\psi_{2}x\right)}^{\infty }e^{-\left(\frac{1}{\lambda_{SPu}}x+\frac{1}{\lambda_{SR}}y\right)}e^{-\frac{1}{\lambda_{SE}}\zeta_{1}}dxdy,\\ \zeta_{1}=\frac{\left(c\lambda_{SE}+\rho^{2}x\right)\left(ay-\phi \left(by+c\lambda_{SR}+\rho^{2}x\right)\right)}{\left[\phi b+(\phi +1)a\right]\left[by+c\lambda_{SR}+\rho^{2}x\right]-aby}, \end{array}$$


Proof: See Appendix A.
(27)$$\mathrm{Pr}1.2=\left\{\begin{array}{cc} 0 & a\leq \theta b\\ \frac{\left(1/\lambda_{RPu}\right)e^{-\psi_{12}/\lambda_{1}}}{\left(1/\lambda_{RPu}\right)+\psi_{2}/\lambda_{1}}-\left(1/\lambda_{1}\lambda_{RPu}\right)I_{2} & a>\theta b \end{array}\right.$$
where
$$\begin{array}{c} \psi 12=\frac{\phi c\lambda_{1}}{\left(a-\phi b\right)};\;\psi 2=\frac{\phi \rho^{2}}{\left(a-\phi b\right)}\;\\ I_{2}=\int_{0}^{\infty }\int_{\left(\psi_{12}+\psi_{2}x\right)}^{\infty }e^{-\left(\frac{1}{\lambda_{RPu}}x+\frac{1}{\lambda_{1}}y\right)}e^{-\frac{1}{\lambda_{RE}}\zeta_{2}}dxdy,\\ \zeta_{2}=\frac{\left(c\lambda_{RE}+\rho^{2}x\right)\left(ay-\phi \left(by+c\lambda_{1}+\rho^{2}x\right)\right)}{\left[\phi b+(\phi +1)a\right]\left[by+c\lambda_{1}+\rho^{2}x\right]-aby}, \end{array}$$

Proof: See Appendix B.

From formulas in (26) and (27), the secrecy outage probability of the U${}_{1}$ is obtained as
(28)$$OP_{U_{1}}=\left\{\begin{array}{cc} 1 & a\leq \phi b\\ 1-\left(\begin{array}{c} \left(\frac{\left(1/\lambda_{SPu}\right)e^{-\psi_{11}/\lambda_{SR}}}{\left(1/\lambda_{SPu}\right)+\left(\psi_{2}/\lambda_{SR}\right)}-\left(1/\lambda_{SR}\lambda_{SPu}\right)\times I_{1}\right)\\ \times \left(\frac{1/\lambda_{RPu}e^{-\psi_{12}/\lambda_{1}}}{1/\lambda_{RPu}+\psi_{2}/\lambda_{1}}-\left(1/\lambda_{1}\lambda_{RPu}\right)\times I_{2}\right)\;\;\; \end{array}\right) & a>\phi b \end{array}\right.$$

### 3.2. Secrecy Outage Probability of U_2_

Similar to U${}_{1}$, the SOP of U${}_{2}$ can be expressed as
(29)$$\begin{array}{cc} OP_{U_{2}} & =\mathrm{Pr}\left[\min\left(SC_{SR}^{e_{2}},SC_{RU_{2}}^{e_{2}}\right)<SC_{th}\right]=1-\mathrm{Pr}\left[SC_{SR}^{e_{2}}\geq SC_{th},SC_{RU_{2}}^{e_{2}}\geq SC_{th}\right]. \end{array}$$

**Proposition** **2.**
*The secrecy outage probability of U*
${}_{2}$
*in (26) is given as*
(30)$$OP_{U_{2}}=1-\left(1-\frac{1}{\lambda_{SPu}\lambda_{SE}}\times I_{3}\right)\;\;\;\times \;\;\;\left(1-\frac{1}{\lambda_{RPu}\lambda_{RE}}I_{4}\right)\;$$
*where*

$\zeta_{3}=\left(\frac{\phi \left(c\lambda_{SR}+\rho^{2}x\right)}{b}+\left(\phi +1\right)\frac{\left(c\lambda_{SR}+\rho^{2}x\right)}{\left(c\lambda_{SE}+\rho^{2}x\right)}y\right),$

$\zeta_{4}=\left(\frac{\phi \left(c\lambda_{2}+\rho^{2}x\right)}{b}+\left(\phi +1\right)\frac{\left(c\lambda_{2}+\rho^{2}x\right)}{\left(c\lambda_{RE}+\rho^{2}x\right)}y\right),$

$I_{3}=\int_{0}^{\infty }\int_{0}^{\infty }\left[e^{-\left(\frac{1}{\lambda_{SPu}}x+\frac{1}{\lambda_{SE}}y\right)}\times \left(1-e^{-\frac{\zeta_{3}}{\lambda_{SR}}}\right)\right]dxdy,$

$I_{4}=\int_{0}^{\infty }\int_{0}^{\infty }\left[e^{-\left(\frac{1}{\lambda_{RPu}}x+\frac{1}{\lambda_{RE}}y\right)}\times \left(1-e^{-\frac{\zeta_{4}}{\lambda_{2}}}\right)\right]dxdy.$



Proof: See Appendix C.

The integrals $I_{1}$ and $I_{2}$ in (28) and $I_{3}$ and $I_{4}$ in (30) are complex integrals and are difficult to resolve practically. In this paper, however, the value of $I_{1},I_{2},I_{3}$ and $I_{4}$ can be found using numerical methods.

## 4. Simulation Results

In this section, the secrecy performance of a NOMA scheme and the impact of imperfect CSI in a UCCN were examined, analyzed, and evaluated. The theoretical results of the analyses were verified with Monte Carlo simulations. The coordinates of S, R, U${}_{1}$, U${}_{2}$, Pu, and E were set to $S\left(0,0\right),R\left(x_{R},0\right.$), $U_{1}\left(x_{U_{1}},y_{U_{1}}\right)=\left(1,0\right)$, $U_{2}\left(x_{U_{2}},y_{U_{2}}\right)=\left(0.75,-0.5\right)$, $Pu\left(x_{Pu},y_{Pu}\right.)$, $E\left(x_{E},y_{E}\right.)$, respectively, in the two-dimensional plane and satisfying $\left(x_{i}>0\right.)$. Hence, $l_{SR}=x_{R}$, $l_{RU_{1}}=x_{U_{1}}-x_{R}$, $l_{RU_{2}}=\sqrt{y_{U_{2}}^{2}+{\left(x_{U_{2}}-x_{R}\right.)}^{2}}$, $l_{RPu}=\sqrt{y_{Pu}^{2}+{\left(x_{Pu}-x_{R}\right.)}^{2}}$, $l_{RE}=\sqrt{y_{E}^{2}+{\left(x_{E}-x_{R}\right.)}^{2}}$, $l_{SE}=\sqrt{y_{E}^{2}+x_{E}^{2}}$, and $l_{SPu}=\sqrt{x_{Pu}^{2}+y_{Pu}^{2}}$. We assume that the target secrecy capacity $SC_{th}=0.5$ (bit/s/Hz) and the exponent β is set to a constant β=3.

Figure 2 and Figure 3 graph the SOP of the two Users U${}_{1}$ and U${}_{2}$ via SNR (dB) with $SC_{th}=0.5$ (bit/s/Hz). The relay R, Pu, U${}_{1}$, U${}_{2}$, and eavesdropper E are located in positions $R\left(x_{R},0\right)=\left(0.5,0\right)$, $\mathrm{Pu}\left(x_{Pu},y_{Pu}\right)=\left(0.5,-1\right)$, $U_{1}\left(x_{U_{1}},y_{U_{1}}\right)=\left(1,0\right)$, $U_{2}\left(x_{U_{2}},y_{U_{2}}\right)=\left(0.75,-0.5\right)$, $E\left(x_{E},y_{E}\right)=\left(0.5,1\right)$, respectively. From the results in Figure 2, we can see the effect of the eavesdropping node E to the SOP when SNR is changed from 0 dB to 20 dB. With ρ=0.95, the SOP values of User U${}_{1}$ are greater than User U${}_{2}$ when SNR < 2.5 dB. Nevertheless, when the SNR increases from 2.5 dB to 30 dB, the SOP of User U${}_{2}$ is better than User U${}_{1}$, and both also increase when the SNR increases as a result of large transmitting power. Besides that, it is noted that imperfect CSI degrades the SOP of the signal.

In Figure 3, we observe the obvious affection of the channel estimation coefficient ρ to the SOP. The SOP of the two users in case ρ=0.95 outperforms the SOP in case ρ=0.9. It means that the system has been impacted by imperfect CSI. We also can see that the secrecy performance of the two Users also decreases when the SNR increases. The security system will be better and it is difficult for the eavesdropper to wiretap the signal. These theoretical results match the simulation results of the proposed system well. Hence, the derived equations are sufficiently accurate for use in analysis.

Figure 4 graphs the SOP of the two Users U${}_{1}$ and U${}_{2}$ via ρ when SNR = 10 (dB) with $SC_{th}=0.5$ (bit/s/Hz). The relay R, Pu, U${}_{1}$, U${}_{2}$, and eavesdropper E are located in positions $R\left(x_{R},0\right)=\left(0.5,0\right)$, $\mathrm{Pu}\left(x_{Pu},y_{Pu}\right)=\left(0.5,-1\right)$, $U_{1}\left(x_{U_{1}},y_{U_{1}}\right)=\left(1,0\right)$, $U_{2}\left(x_{U_{2}},y_{U_{2}}\right)=\left(0.75,-0.5\right)$, $E\left(x_{E},y_{E}\right)=\left(0.5,1\right)$, respectively. As observed in Figure 4, the secrecy performance of U${}_{1}$ is better than U${}_{2}$ when ρ<0.9. However, when the correlation coefficient ρ>0.9, the SOP of U${}_{2}$ is less than U${}_{1}$. This means that the effects of the evaluation errors decrease when the correlation coefficients ρ increase. In addition, the secrecy performance of two Users is more efficient when the channel estimation coefficient ρ is higher.

Figure 5 graphs the SOP of the proposed scheme versus the position of the of eavesdropper E on the *y*-axis when the coordinate value $y_{E}$ changes from 0.2 to 2 when SNR = 10 (dB), (bit/s/Hz), $R\left(x_{R},0\right)=\left(0.5,0\right)$, $\mathrm{Pu}\left(x_{Pu},y_{Pu}\right)=\left(0.5,-1\right)$, $U_{1}\left(x_{U_{1}},y_{U_{1}}\right)=\left(1,0\right)$, $U_{2}\left(x_{U_{2}},y_{U_{2}}\right)=\left(0.75,-0.5\right)$. Figure 5 shows that the SOP of U${}_{1}$ and U${}_{2}$ decrease when $y_{E}$ increases. This means that the secrecy transmission of the two Users will be intact when eavesdropper E moves farther away from source S and relay R.

We can also see that the secrecy performance of U${}_{2}$ is better than U${}_{1}$ as a result of the proposed scheme applying NOMA and using SIC to detect the signal under the effect of imperfect CSI on the system.

Figure 6 graphs the SOP of User U${}_{1}$ and U${}_{2}$ versus the power allocation coefficients $\beta_{1}$ (changing from 0.6 to 0.95) when SNR = 10 (dB), $SC_{th}=0.5$ (bit/s/Hz), $R\left(x_{R},0\right)=\left(0.5,0\right)$, $\mathrm{Pu}\left(x_{Pu},y_{Pu}\right)=\left(0.5,-1\right)$, $U_{1}\left(x_{U_{1}},y_{U_{1}}\right)=\left(1,0\right)$, $U_{2}\left(x_{U_{2}},y_{U_{2}}\right)=\left(0.75,-0.5\right)$, $E\left(x_{E},y_{E}\right)=\left(0.5,1\right)$. This figure shows the impact of a varying β1 on the system. When $\beta_{1}$ increases from 0.6 to 0.85, the SOP of U${}_{2}$ outperforms the SOP of U${}_{1}$. The secrecy transmission of U${}_{1}$ is then better than U${}_{2}$ when $\beta_{1}>0.85$. We can thus observe the impact of $\beta_{1}$ on the security performance of a UCCN system with a NOMA solution under the effect of imperfect CSI.

## 5. Conclusions

A NOMA scheme with imperfect CSI in a UCCN was proposed in this paper. We also researched the physical layer security to improve the secrecy performance. The secrecy performance was examined analyzed, evaluated by the secrecy outage probability of the achievable secrecy capacity, and over Rayleigh fading channels. The obtained results show that the security performance of the system model is decreased when the imperfect CSIs exists as well as the SNR increases. The proposed scheme with the optimal imperfect CSIs can achieve the best performance. Besides that, we can see that the node eavesdropper E is far from the source S and relay R, the security performance became more security. In addition, the security performance of U1 and the security performance of U2 are also compared with together. Lastly, the achieved results of the SOP matched well with the Monte Carlo simulation results.

## Figures and Tables

**Figure 1 sensors-20-00895-f001:**
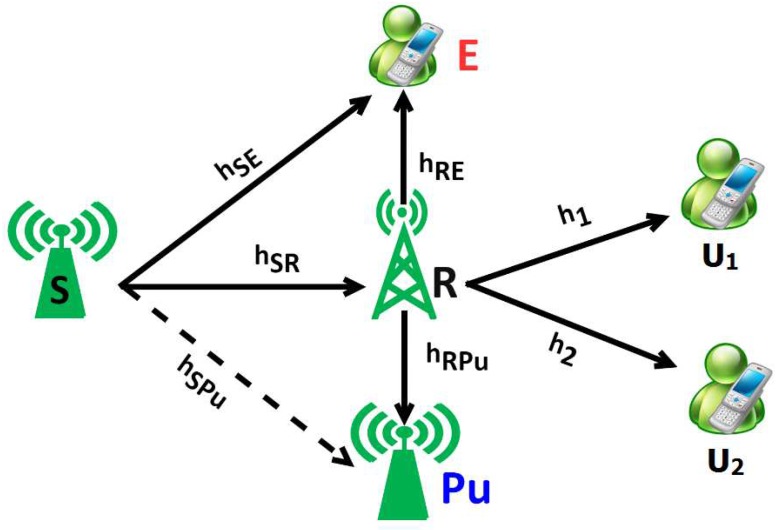
System model of NOMA and PLS under imperfect CSI in a UCCN.

**Figure 2 sensors-20-00895-f002:**
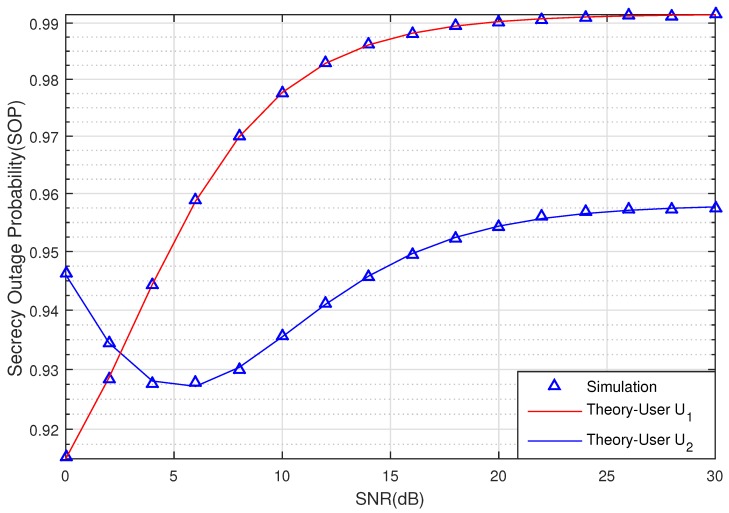
The SOP of U${}_{1}$ and U${}_{2}$ versus SNR (dB).

**Figure 3 sensors-20-00895-f003:**
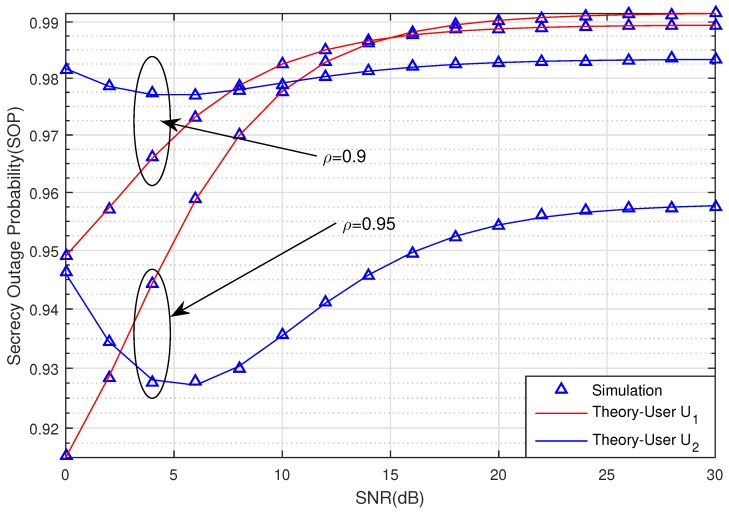
The SOP of U${}_{1}$ and U${}_{2}$ versus SNR (dB) when ρ=0.9 and ρ=0.95.

**Figure 4 sensors-20-00895-f004:**
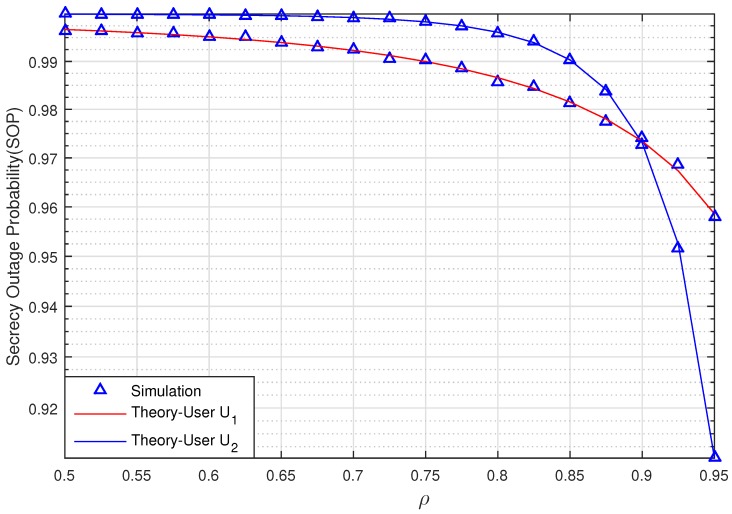
The SOP of U${}_{1}$ and U${}_{2}$ via ρ when SNR = 10 (dB).

**Figure 5 sensors-20-00895-f005:**
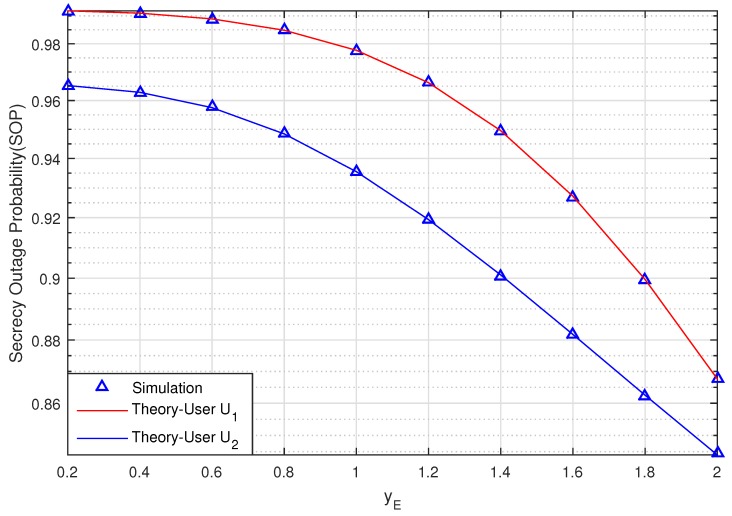
The SOP of U${}_{1}$ and U${}_{2}$ versus coordinate $y_{E}$ of eavesdropper E when SNR = 10 (dB).

**Figure 6 sensors-20-00895-f006:**
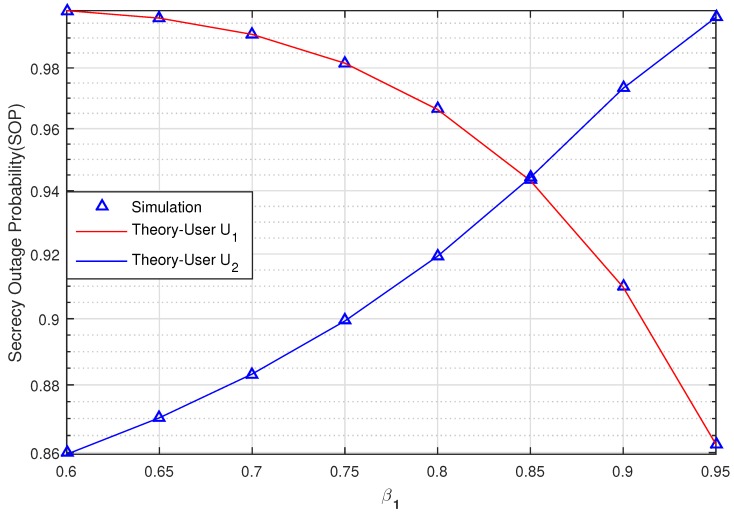
The SOP of U${}_{1}$ and U${}_{2}$ versus the power allocation coefficients $\beta_{1}$ when SNR = 10 (dB).

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
