# Peer review of "Secrecy Outage Probability of a NOMA Scheme and Impact Imperfect Channel State Information in Underlay Cooperative Cognitive Networks"

_sensors, 2020, doi:10.3390/s20030895_

Round 1
Reviewer 1 Report
Security performance and the impact of imperfect channel state 1 information (CSI) in 2 underlay cooperative cognitive networks (UCCN) is investigated in this paper. The comments are listed as follows
the most important thing for CR is spectrum sensing and access, the authors should express why do the study security. how to decrease the interference between NOMA users, CR and PU. how to judge wether the user is securaty. please give the comparison between Security outage and common outage. some relative work on NOMA CR can be described such as
A Novel Multichannel Internet of Things Based on Dynamic Spectrum Sharing in 5G Communication," IEEE Internet of Things Journal, vol. 6, no. 4, pp. 5962-5970, Aug. 2019.
NOMA-based Resource Allocation for Cluster-based Cognitive Industrial Internet of Things. IEEE Transactions on Industrial Informatics., online, 2019.
Author Response
Reviewer1's comments to the authors:
Comment #1: Security performance and the impact of imperfect channel state 1 information (CSI) in 2 underlay cooperative cognitive networks (UCCN) is investigated in this paper. The comments are listed as follows the most important thing for CR is spectrum sensing and access, the authors should express why do the study security. how to decrease the interference between NOMA users, CR and PU. how to judge whether the user is security. please give the comparison between Security outage and common outage. some relative work on NOMA CR can be described such as:
A Novel Multichannel Internet of Things Based on Dynamic Spectrum Sharing in 5G Communication," IEEE Internet of Things Journal, vol. 6, no. 4, pp. 5962-5970, Aug. 2019. NOMA-based Resource Allocation for Cluster-based Cognitive Industrial Internet of Things. IEEE Transactions on Industrial Informatics., online, 2019.
Response #1: The authors would like to thank the Reviewer for the suggestions.
- In the revised version, we added “why we deal with the physical layer security” as well as explanations about NOMA users, CR and PU in the introduction section.
- We also added the next related references [1-2]
- “Please give the comparison between Security outage and common outage”: In order to calculate the outage probability, we usually use the value of SNR and it is set smaller than a threshold. In our article, we investigated the PLS, so the security outage probability is based on the achievable secrecy capacity and the approach is used also in other papers such as [3], [4].
[1] A Novel Multichannel Internet of Things Based on Dynamic Spectrum Sharing in 5G Communication," IEEE Internet of Things Journal, vol. 6, no. 4, pp. 5962-5970, Aug. 2019.
[2]NOMA-based Resource Allocation for Cluster-based Cognitive Industrial Internet of Things. IEEE Transactions on Industrial Informatics., online, 2019.
[3] Chen, J., Yang, L. and Alouini, M.S. Physical Layer Security for Cooperative NOMA Systems. IEEE 281 Transactions on Vehicular Technology 2018, 67, 4645-4649.
[4] Duy, T.-T.; Duong, T.-Q.; Thanh, T.-L.; Bao, V.N.-Q. Secrecy performance analysis with relay selection 272 methods under impact of co-channel interference. IET Communications 2015, 9, 1427-1435.

Reviewer 2 Report
The article presented by the authors is a bit hard to be judged. In the introduction general description of the scientific problem was described - the impact of imperfect channel state information in the UCCN.
In the 2nd part of the article authors analyzed the secrecy outage probability. In the reviewer point of view this part is the biggest advantage and disadvantage at the same time. The form of presentation (many formulas and mathematical notation was used) is influenced by the scientific and deep analysis of the problem but unfortunately it is very hard to follow and understand for the reader. In the reviewer point of view this section must be modified - better and wider described.
In the cessation containing the simulation results the main assumptions were not explained enough. It is also advised to add a wider comment to the obtained results (also extend the conclusions section).
In the reliever point of view the article needs to be modified to present the idea in a clearer way to the reader.
Author Response
Reviewer2's comments to the authors:
Comment #1: The article presented by the authors is a bit hard to be judged. In the introduction general description of the scientific problem was described - the impact of imperfect channel state information in the UCCN.
-In the 2nd part of the article authors analyzed the secrecy outage probability. In the reviewer point of view this part is the biggest advantage and disadvantage at the same time. The form of presentation (many formulas and mathematical notation was used) is influenced by the scientific and deep analysis of the problem but unfortunately it is very hard to follow and understand for the reader. In the reviewer point of view this section must be modified - better and wider described.
Response #1: The authors would like to thank the Reviewer for the helpful comments. In the section 2, section 3 as well as Appendix part, we have modified and described mathematical formulas in the revised version to be more understandable.
Comment #2: In the cessation containing the simulation results the main assumptions were not explained enough. It is also advised to add a wider comment to the obtained results (also extend the conclusions section).
Response #2: In the revised version, we rewrote the simulation results. In addition, we also added the extension in the section CONCLUSIONS. (text in blue)

Reviewer 3 Report
The authors evaluate the secrecy performance, in terms of outage probability, of underlay cooperative cognitive networks based on non-orthogonal multiple access channels with imperfect channel state information. Unfortunately, I have several concerns that makes the work unpublishable in Sensors. Although I don’t perfectly catch up the calculation performed in the manuscript, I believe it does not have enough innovation and advancement for the journal. In the case of resubmission to any other journals, heavy modifications will be required.
Below is my concern:
1) The main concern is about novelty of this manuscript. The abstract says “Through numerical analysis, the results showed that the proposed system’s secrecy performance became more efficient when the eavesdropper node E was farther away from the source node S and the intermediate cooperative relay R.” To my knowledge, this is common characteristics in any physical layer security schemes. So, this is too weak for the main claim of the manuscript.
2) Introduction contains a lot of technical terms and abbreviations. However, none of them are explicitly defined in introduction. This makes the manuscript definitely difficult especially for the readers unfamiliar with the wireless communications technologies. Some of the abbreviations are defined in Abstract and Key word section, of which some readers miss them. Consequently, this degrades the readability of the manuscript.
3) The advancement of the present scheme to existing ones is not explained. For example, in Introduction, the Authors explains the benefit of the combination of NOMA and PLS in a UCCN. The numerical evaluations only explain the fundamental characteristics of the present scheme, which should have contained the comparison of the present scheme with existing schemes. These points degrade the novelty of the manuscript.
4) Figure 1 does not yield much information in my opinion. This is because the node PU is introduced without any definition.
Author Response
Reviewer3's comments to the authors:
Comment #1: 1) The main concern is about novelty of this manuscript. The abstract says “Through numerical analysis, the results showed that the proposed system’s secrecy performance became more efficient when the eavesdropper node E was farther away from the source node S and the intermediate cooperative relay R.” To my knowledge, this is common characteristics in any physical layer security schemes. So, this is too weak for the main claim of the manuscript.
Response #1: We would like to thank the Reviewer for the helpful and constructive comments.
In our article, we researched not only physical layer security(PLS), non-orthogonal multiple access (NOMA) but also studied the impact of imperfect channel state information (CSI) in underlay cooperative cognitive networks (UCCN). This issue we consider as important in wireless communications and its analysis is difficult and complex.
Comment #2: Introduction contains a lot of technical terms and abbreviations. However, none of them are explicitly defined in introduction. This makes the manuscript definitely difficult especially for the readers unfamiliar with the wireless communications technologies. Some of the abbreviations are defined in Abstract and Key word section, of which some readers miss them. Consequently, this degrades the readability of the manuscript.
Response #2: In revised version, we added and described the technical terms main such as cognitive radio(CR), physical layer security(PLS), and non-orthogonal multiple access (NOMA) in the introduction section (line 16-25, line 31-34).
Comment #3: The advancement of the present scheme to existing ones is not explained. For example, in Introduction, the Authors explains the benefit of the combination of NOMA and PLS in a UCCN. The numerical evaluations only explain the fundamental characteristics of the present scheme, which should have contained the comparison of the present scheme with existing schemes. These points degrade the novelty of the manuscript.
Response #3: In our paper, we only compare the secrecy performance between U1 and U2 in the combination of NOMA and PLS in a UCCN under the impact of imperfect channel state information (CSI).
Comment #4: Figure 1 does not yield much information in my opinion. This is because the node PU is introduced without any definition.
Response #4: In the revised version, we added and described clearly the node PU in the system model section (line 84-86) as well as in the introduction about Cognitive radio (line 15-19)

Round 2
Reviewer 1 Report
the paper has addressed all my problems now it can be accepted
Reviewer 3 Report
I appreciated authors' efforts to improve the manuscript. Publichation of the paper would be okay, unless other reviewers disagree.